# Cardiopulmonary Arrest Caused by Large Substernal Goiter—Treatment with Combined Cervical Approach and Median Mini-Sternotomy: Report of a Case

**DOI:** 10.3390/medicina57040303

**Published:** 2021-03-24

**Authors:** Charilaos Koulouris, Aristoklis Paraschou, Vasiliki Manaki, Stylianos Mantalovas, Kassiani Spiridou, Andreana Spiridou, Styliani Laskou, Nickos Michalopoulos, Petru Adrian Radu, Dan Cartu, Valeriu Șurlin, Victor Strambu, Isaak Kesisoglou, Konstantinos Sapalidis

**Affiliations:** 13rd Surgery Department, Medical School of Health Sciences, AHEPA University Hospital, Aristotle University of Thessaloniki, 54636 Thessaloniki, Greece; charilaoskoulouris@gmail.com (C.K.); aristoklisparaschou@gmail.com (A.P.); vassiamanaki@gmail.com (V.M.); steliosmantalobas@yahoo.gr (S.M.); kasiani18@hotmail.com (K.S.); andreana10_@windowslive.com (A.S.); stelaskou@gmail.com (S.L.); nickos.michalopoulos@gmail.com (N.M.); ikesis@hotmail.com (I.K.); 2Department of Surgery, University of Medicine and Pharmacy Carol Davila Bucharest, 020021 București, Romania; drradupetru@yahoo.com (P.A.R.); drstrambu@gmail.com (V.S.); 31st Department of Surgery, University of Medicine and Pharmacy of Craiova, 200349 Craiova, Romania; cartu_dan@hotmail.com (D.C.); vsurlin@gmail.com (V.Ș.)

**Keywords:** goiter, thyroidectomy, sternotomy

## Abstract

Introduction: Substernal goiter is usually defined as a goiter that extends below the thoracic inlet or a goiter with more than 50% of its mass lying below the thoracic inlet. Substernal goiters may compress adjacent anatomical structures causing a variety of symptoms. Case report: Here we report a rare case of a 75-year-old woman presenting with cardiac arrest caused by acute respiratory failure due to tracheal compression by a substernal goiter. Discussion: Substernal goiters can be classified as primary or secondary depending on their site of origin. Symptoms are diverse and include a palpable neck mass, mild dyspnea to asphyxia, dysphagia, dysphonia, and superior vena cava syndrome. Diagnosis of substernal goiter is largely based on computed tomography imaging, which will show the location of the goiter and its extension in the thoracic cavity. Surgery is the treatment of choice for symptomatic patients with substernal goiter. The majority of substernal goiters are resected through a cervical approach. However, in approximately 5% of patients, a thoracic approach is required. The most important factor determining whether a thoracic approach should be used is the depth of the extension to the tracheal bifurcation on CT imaging. Conclusion: Cardiac arrest appearing as the first symptom of a substernal goiter is a very rare condition and should be treated by emergency thyroidectomy via a cervical or thoracic approach depending on the CT imaging findings.

## 1. Introduction

Substernal goiter is generally defined as a goiter that extends below the thoracic inlet or a goiter that lies below the thoracic inlet with more than 50% of its mass. The prevalence of such goiters ranges from 2–19% of confirmed thyroidectomies [1]. Substernal goiters can cause various symptoms due to the compression of adjacent anatomical structures. Airway compression can cause mild dyspnea to extreme asphyxia, oesophageal compression may occur as dysphagia, while vascular part compression may cause vascular symptoms [1].

We report herein a rare case of a 75-year-old woman presenting with cardiac arrest caused by acute respiratory failure due to tracheal compression by a substernal goiter.

## 2. Case Report

A 75-year old female was admitted to the emergency department of our hospital (AHEPA University Hospital of Thessaloniki) with cardiac arrest due to acute respiratory failure. Immediate cardiopulmonary resuscitation was performed accompanied by copious tracheal intubation and respiratory support. The patient was then transferred to the intensive care unit.

Physical examination revealed a barely palpable goiter. Laboratory tests at normal admission are mentioned in Table 1. Computed tomography (CT) revealed a large nodular substernal goiter, which repelled the large vessels and compressed the trachea and the oesophagus to the left (Figure 1 and Figure 2). Goiter dimensions were 9.3 cm × 9.4 cm × 5.4 cm.

On the second day of hospitalization in ICU, the patient was extubated but soon showed signs of airway obstruction. She was again intubated. Due to lack of medical evidence for other causes of cardiopulmonary arrest, the indication of total thyroidectomy was set so as to decompress the adjacent structures. The large size and the inability to perform safe mobilization of the goiter led us to perform a Kocher incision plus a median mini-sternotomy. (Figure 1, Figure 2, Figure 3 and Figure 4). Intraoperative tracheostomy was also performed due to tracheomalacia caused by chronic compression to the neck (Figure 3, Figure 4, Figure 5 and Figure 6). Tracheomalacia was observed due to long-term tracheal compression.

Postoperatively the patient stayed in the ICU; on the 18th postoperative day she was transferred to the ward. On the 38th postoperative day, the tracheostomy was removed (Figure 5 and Figure 7). The patient progressively retained sufficient oxygenation without the need for oxygen supply. She was discharged on the 45th postoperative day to a rehabilitation center.

Regarding the histopathology report, typical lesions of chronic lymphocytic thyroiditis were recognized.

## 3. Discussion

Depending on their site of origin, substernal goiters may be categorized as primary or secondary. Primary substernal goiters (less than 1%) derive from ectopic thyroid tissue within the mediastinum, whereas secondary substernal goiters evolve due to cervical thyroid tissue’s downward development. An instance of secondary substernal goiter is the present case study. Substernal goiters are typically discovered during the fifth and sixth decades of life because of their slow growth, with the female:male ratio being 3:1 [1].

Most of the patients have a palpable neck mass [2]. Compression of mediastinal structures, such as the trachea, the broad arteries and the esophagus, results in the most frequently presented symptoms. Tracheal compression symptoms may range from a mild cough or dyspnea to severe asphyxia [3]. In our case, cardiac arrest was caused by life-threatening airway compression. Intubation as an emergency procedure for respiratory distress is very rare, accounting for only 6% of cases with substernal goiter. Dysphagia, dysphonia, superior vena cava syndrome, and signs of hyperthyroidism are among other symptoms. More unusual symptoms are syncopal episodes due to left vertebral artery compression and recurrent pulmonary emboli along with superior vena cava syndrome [4,5].

Diagnosis of substernal goiter is mainly dependent on imaging techniques. Chest radiograph may indicate tracheal displacement and may successfully imply the presence of a substernal goiter, but computed tomography (CT) scan is of vital importance for the preoperative assessment of the position of the goiter and its extension in the thoracic cavity [6].

Surgery is the recommended treatment for symptomatic patients with substernal goiter [7]. Medical treatment is not recommended because of its unpredictable and usually short-lived results [8]. Preventive surgery can also be performed in asymptomatic patients in order to prevent potential compression [7]. The risk of malignancy for substernal goiters is between 3–21% [9]. The majority of substernal goiters are resected through a cervical approach. However, a thoracic approach is needed in approximately 5% of patients. Thoracic approach may include partial or total sternotomy, video-assisted thoracic surgery, and lateral thoracotomy [10]. These techniques can be used alone or combined. The combination of cervical incision and partial sternotomy has been carried out in our case (cervico-sternotomy). Cervico-sternotomy has an excellent outcome in large substernal goiters that are closely linked to mediastinal structures, as it enables the complete removal of the mass of the thyroid from the cervical area and mediastinum and the healthy decompression of the trachea [11]. The depth of the extension to the tracheal bifurcation on CT imagery is the most significant factor determining whether a thoracic approach should be used. Extension of goiter into the posterior mediastinum, dumbbell form, or wider thoracic than the thyroid portion may be additional findings of the CT. Other indications for a thoracic approach include ectopic goiter, invasive carcinoma, and goiter reoperation [10]. The authors also emphasize that the decision to perform a sternotomy is always made during the procedure.

Transient hypoparathyroidism is the most frequent complication following surgery (15%). Other post-opoperative complications include transient recurrent laryngeal nerve (RLN) injury (12%), permanent RLN injury (4.3%), postoperative hemorrhage (1%), subcutaneous hematoma or seroma (3%), and wound infection (1.5%) (10]. Tracheomalacia is a usual complication of substernal goiter caused by long-term tracheal compression [3]. In our case, it was handled by performing intraoperative tracheostomy. In a study of 19,662 patients undergoing total thyroidectomy, those with substernal goiter treated with a cervical approach had increased risk for postoperative transient and permanent hypoparathyroidism, transient and permanent monolateral, and permanent bilateral RLN palsies compared to patients with cervical goiter. However, patients with substernal goiter treated with a sternal split approach had an increased risk only for transient hypoparathyroidism and transient bilateral RLN palsy compared to patients with cervical goiter. These findings suggest that a sternal split approach should be considered in the presence of a broad substernal goiter, since it can provide the surgeon with greater control of all the adjacent structures, taking into account the severe complications of the mediastinum, such as mediastinitis [12].

Only seven articles in the literature refer to compression of adjacent organs due to a substernal goiter [13,14,15,16,17,18,19]. In five of them, the reported patients were presented with cardiopulmonary arrest [13,14,15,16,17]. Two of the five cases occurred during pregnancy [14,15]. Total or partial thyroidectomy was the reported treatment.

## 4. Conclusions

In conclusion, cardiac arrest appearing as the first symptom of a substernal goiter is a very rare condition and it should be treated with emergency thyroidectomy via a cervical or thoracic approach depending on the CT imaging and the intraoperative findings. Diagnosis is really challenging and occurs after investigation and exclusion of the other possible causes of cardiopulmonary arrest.

## Figures and Tables

**Figure 1 medicina-57-00303-f001:**
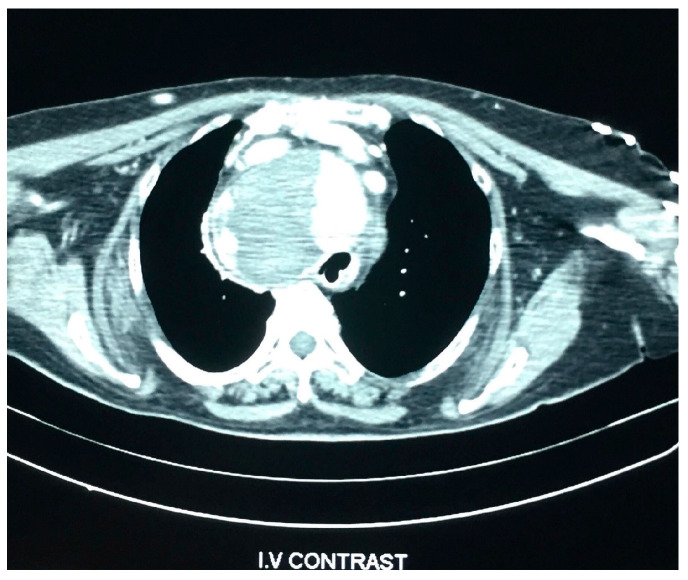
CT scan showing the enlarge thyroid gland and the compressed trachea shifted to the left.

**Figure 2 medicina-57-00303-f002:**
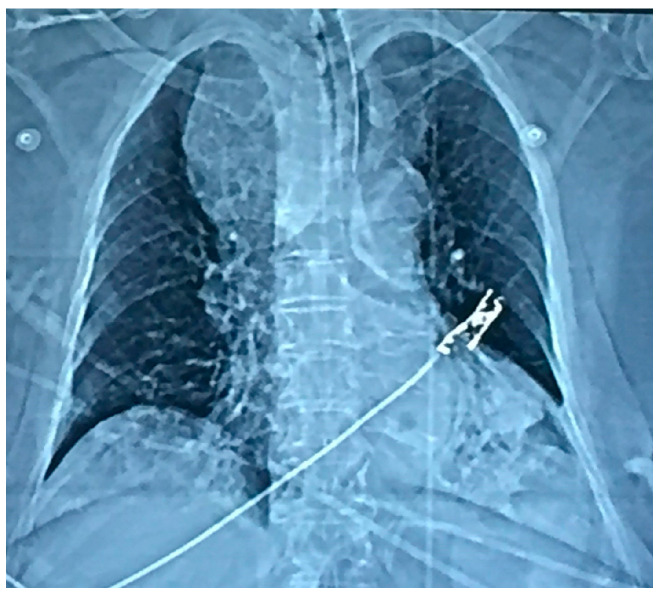
Chest x-Ray of the patient.

**Figure 3 medicina-57-00303-f003:**
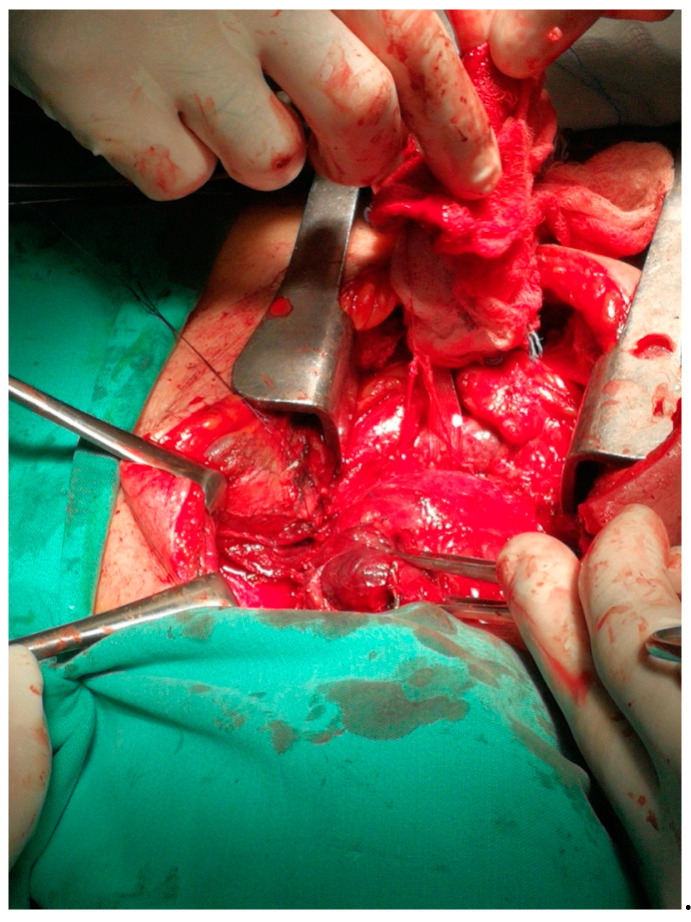
The combination of cervical incision and partial sternotomy (cervico-sternotomy) or the removal of the substernal goiter.

**Figure 4 medicina-57-00303-f004:**
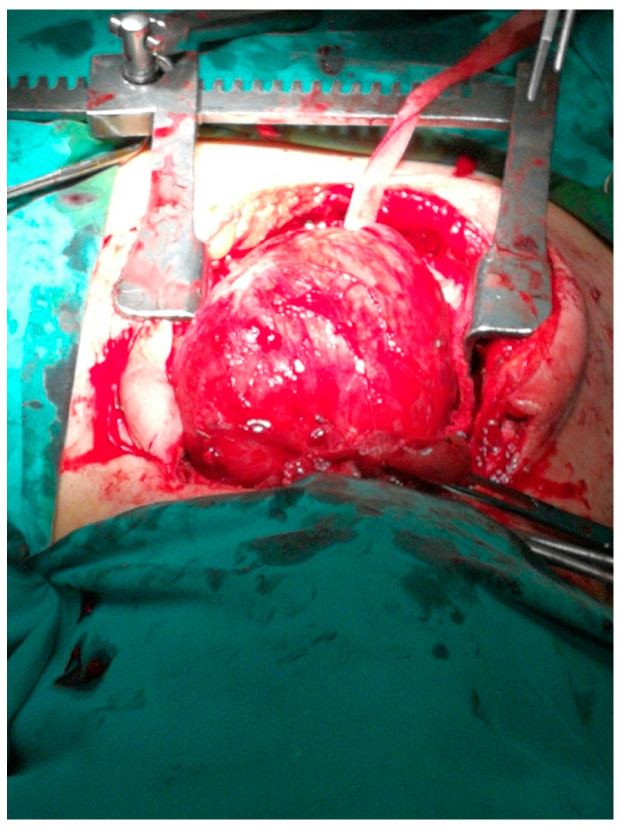
The mobilization of the substernal goiter through the cervical incision and partial sternotomy (cervico-sternotomy).

**Figure 5 medicina-57-00303-f005:**
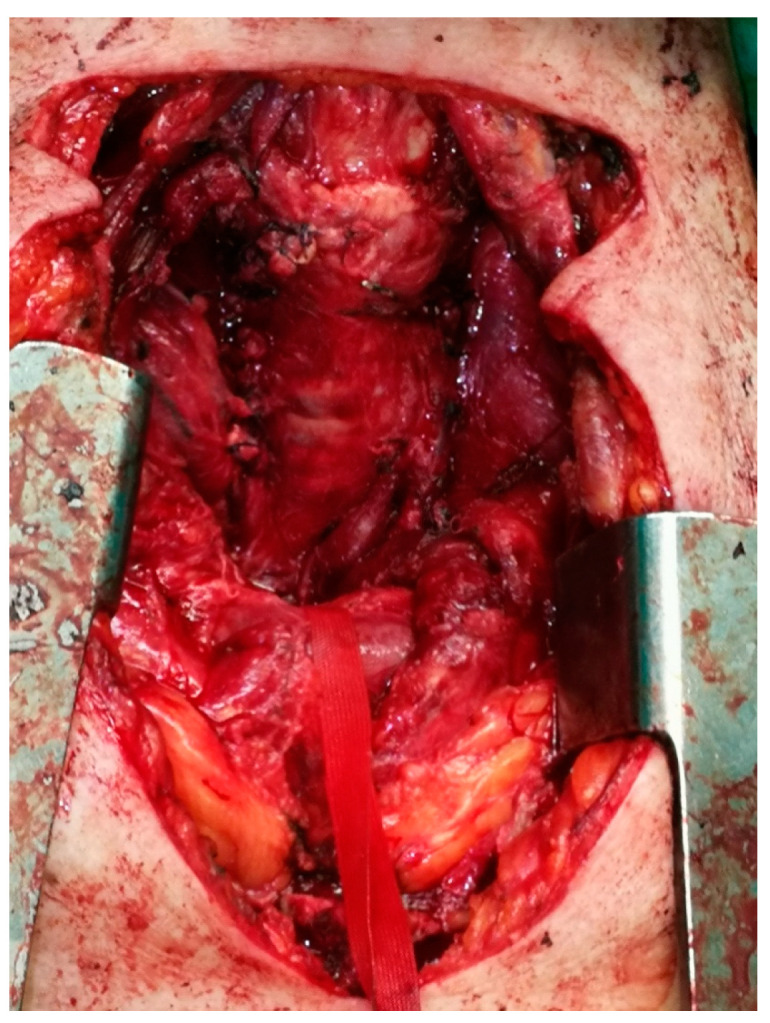
The residual cavity after the surgical removal of the substernal goiter.

**Figure 6 medicina-57-00303-f006:**
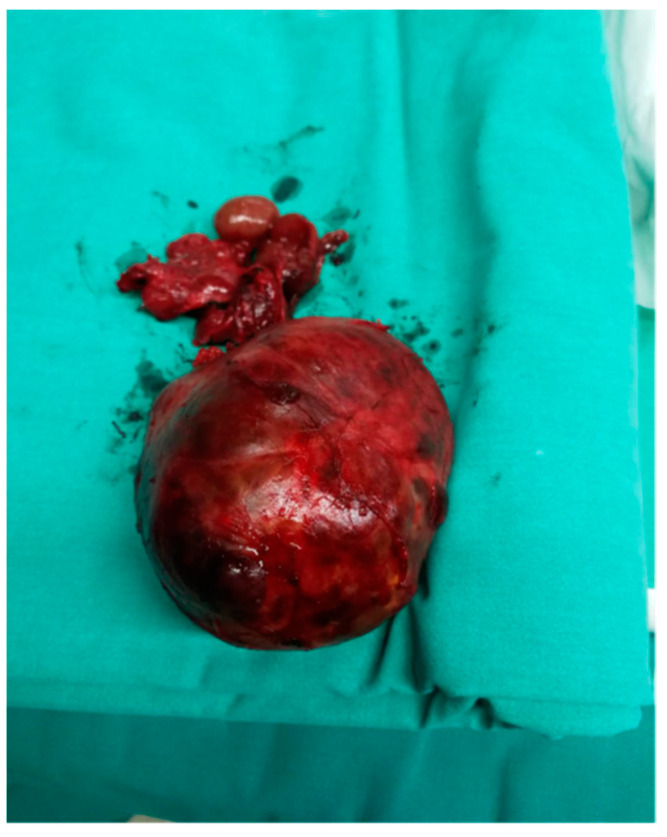
The thyroidectomy specimen.

**Figure 7 medicina-57-00303-f007:**
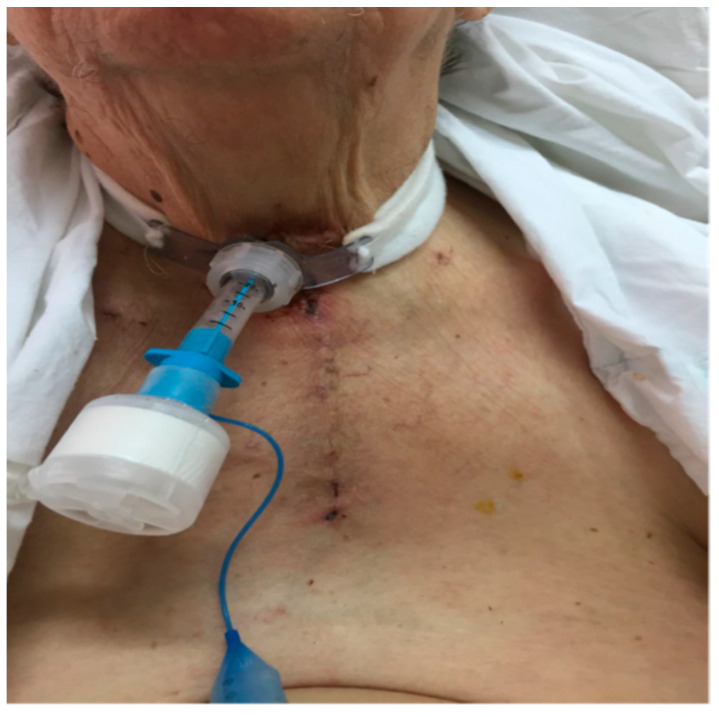
Thirty-eighth postoperative day (POD 38), just before the removal of the tracheostomy tube.

**Table 1 medicina-57-00303-t001:** Biochemical Tests at admission.

Glucose	116 mg/dL
WBC	13.070 × 106/mL
K+	3.8 mmol/l
Na+	142 mmol/l
Troponin THS	11 pg/mL
SGOT	22 U/L
SGPT	10 U/L
CPK	60 U/L
CK-MB	11 U/L
D-Dimers	492 ng/mL

## Data Availability

Bioethics committee approves the request of the Head of Third Department of Surgery of Ahepa University Hospital I. Kesisoglou, concerning data editing of the patient’s medical record for the manuscript entitled “Cardiopulmonary arrest caused by large substernal goiter-treatment with combined cervical approach and median-mini sternotomy” with scientific supervisors K. Sapalidis (Associate Professor of Surgery) and I. Kesisoglou (Head of the Department) to the extent that the GDPR of the patient’s dossier is respected.

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
