# Peer review of "Cardiopulmonary Arrest Caused by Large Substernal Goiter—Treatment with Combined Cervical Approach and Median Mini-Sternotomy: Report of a Case"

_medicina, 2021, doi:10.3390/medicina57040303_

Round 1

Reviewer 1 Report

It is not specified where the clinical case was treated.

The real cause of cardio-respiratory arrest is not clear from the diagnostic-therapeutic process.

The conclusions are too general and unrelated to the discussion.

To make the manuscript more interesting it would be useful to add a review of the literature.

Author Response

Thank you for your rinteresting comments. We performed the suggested modifications to the manuscript. We are looking forward to your revision

Reviewer 2 Report

This paper seems to have a clear argument and its content is well expressed. However, this paper requires minimal English grammar correction, spacing, and period writing. Additionally, please correct whether the reference citation is appropriate, whether there is any omission, or the incorrect marking position.

Question 1) In this paper, why was  total thyroidectomy performed, not lobectomy or subtotal thyroidectomy? Is it because there is a possibility of recurrence or that hidden malignancies cannot be ruled out?

Author Response

Total thyroidectomy was mainly chosen so as to excise all the thyroid parenchyma that compressed the heart and the major thoracic vessels. Secondly, the risk of malignancy could not be excluded

Reviewer 3 Report

Thank you for giving me the opportunity to review your work. The authors reported a case regarding cardiac arrest caused by large substernal goiter. It is very interesting topic, well written manuscript, good figures, well presented with sensible conclusions.

Author Response

Thank you for your rinteresting comments. We performed the suggested modifications to the manuscript. We are looking forward to your revision.

Round 2

Reviewer 1 Report

The authors have included the suggested corrections.